# Cognitive Model for Assessing the Security of Information Systems for Various Purposes

**Vladimir V. Baranov** [1,*] **and Alexander A. Shelupanov** [2,*]

1   Department of "Information Security", M.I. Platov South Russian State Polytechnic University, 346428 Novocherkassk, Russia
2   Department of "Integrated Information Security of Electronic Computing Systems", Tomsk State University of Control Systems and Radioelectronics (TUSUR), 634034 Tomsk, Russia
*   Correspondence: kaf-ib@npi-tu.ru (V.V.B.); saa@tusur.ru (A.A.S.); Tel.: +7-9281-00-05-98 (V.V.B.)

**Abstract:** This article substantiates the relevance of the development of a cognitive model for assessing the security of information systems for various purposes, designed to support decision-making by officials of information security management bodies. The article analyzes scientific papers and research in this area, formulates the requirements for the functional capabilities of the model, and investigates and identifies the most appropriate modeling tools, based on the symmetry property that develops from integrated ontological and neuro-Bayesian models; typical clusters of information systems; tactics and techniques for the implementation of information security threats through the vulnerabilities of objects at various levels of the International Organization of Standardization/Open Systems Interconnection model (the ISO/OSI model); protective influences; and attacking influences. This approach allowed us to identify such objects of influence and their current vulnerabilities and scenarios for the implementation of information security threats; to calculate the joint probability distribution of information security events of various origins; and to simulate the process of operational management of information security.

**Keywords:** cognitive model; ontological model; neuro-Bayesian model; objects of influence; vulnerabilities; tactics; techniques; information protection measures

## 1. Introduction

Ensuring reliable protection of information systems (IS) for various purposes requires the development of a methodological apparatus for modeling the processes of their functioning under the conditions of the complex impacts of various categories of violators. In practice, this task is complicated by probabilistic assessment, incomplete data, and a high degree of uncertainty regarding the nature of violators' actions and their choices of certain ways to implement threats to information security.

The scientific value of this study lies in the fact that the cognitive model of the functioning of IS for various purposes, as presented in this article, under the conditions of the complex impacts, is the next step in the development of the theory and methodology of information security, in terms of creating new, and improving existing, methods, models, and algorithms for decision support in the formation of sets of measures to protect IS for various purposes. For the first time, this research uses methods that integrate ontological structural-functional and probabilistic neuro-Bayesian models (NBMs), the representation of objects of destructive influence (DI) at the levels of the International Organization of Standardization/Open Systems Interconnection model (the ISO/OSI model), the determination of access points to information systems and ways of distributing information security threat (IST) scenarios, the consideration of attack vectors, the determination of the degree of criticality of elements in standard IS modules, and the calculation of numerical values of indicators for evaluating the effectiveness of solutions for the organization of protective

measures. Solutions are also proposed to reduce the subjectivity of expert decisions and to control self-learning systems, based on artificial intelligence.

This paper does not consider algorithms for the software implementation of the developed cognitive model. The methods of determining scenarios and tactics for the implementation of information security threats are conceptually outlined. The appearance of the NBM is also conceptually presented, with directions for further research.

This study has the following structure. Section 2 provides an overview of the literature. Section 3 identifies the main methods used in the study. Section 4 presents the results of the study and discusses and evaluates their effectiveness. Section 5 provides the algorithms used for determining the relevance of various scenarios and tactics. In Section 6, the results are summarized, conclusions are drawn, and suggestions for further research are provided.

## 2. Review of Scientific Literature on the Research Topic

The key aspect in the process of solving the problem considered in the paper is to ensure the required level of symmetry of the results of modeling the risks of the implementation of the ISTs and the actual situation. To develop and evaluate the effectiveness of the solutions, the mathematical apparatus of multi-criteria evaluation (MCDM methods) is used. From a variety of alternative solutions, the most effective option for the current situation is selected according to several criteria. An overview of such methods is provided in [1].

In [2], an original hybrid multicriteria evaluation method, AHP-TOPSIS-2N, was proposed. That method was an integration of the analytical hierarchy process (AHP), the method of order preference by similarity to the ideal solution (TOPSIS), and two normalization procedures (2N).

At the stage of functioning, in the conditions of dynamically changing destructive impacts (DIs), the IS protection system requires stable and continuous operational management, as well as a timely and reasonable response to newly emerging risks and ISTs. Modeling of this process is required. The cognitive model presented in the study for assessing the security of information systems for various purposes served as the basis for creating a decision support system (DSS) for government officials in the field of information security [3].

To solve the problem of developing a cognitive model for assessing the security of information systems for various purposes in conditions of complex impacts and fuzzy initial data, a number of scientific studies in this field were analyzed [4–6]. In this article, we consider the conceptual apparatus and the existing methodological approaches that are applied in the process of developing the model.

ISTs originate from external or internal violators, with the aim of damaging the objects of influence at different levels of IS through their vulnerabilities. ISTs are implemented in various ways. Each method has implementation scenarios containing tactics and techniques [7].

The process of data transmission and storage is displayed by an open network model—the Basic Reference Model Open Systems Interconnection model (the OSI/ISO model)—which has seven levels [8]. With the help of that model, it is easy to determine the structural and functional connections of elements of local information, computing networks, and IS in general. Everything that happens when sending and receiving data, as well as the physical and logical devices involved in those processes, interfaces; Protocols are described in detail by a seven-level OSI/ISO model.

If the objects of violators' impacts on information systems are linked to the OSI/ISO model, then it is possible to distinguish the following levels of their placement: physical, channel, network transport, session, representative, and applied. In order to ensure the completeness of accounting for impact objects, in addition to the above levels, it is proposed to allocate a user level that is not included in the OSI/ISO model. The solution of this problem will reveal the interlevel structural and functional relationships between the objects of the impact of IS (the IS elements).

The objects of influence at different levels implement processes of varying degrees of criticality to ensure the sustainable functioning of IS (the IS elements). Currently, their mutual influence and safety indicators at the integrative level are poorly investigated [9]. The determination of actual ISTs is carried out by the expert method during the construction of the IST model, which also includes the violator model. The degree of danger of a threat is determined by the magnitude of the risk and the degree of damage during its implementation. A realized threat resulting in damage is an incident. Protection against ISTs is carried out by the use of information security measures. Information protection measures (IPMs) are divided into technical measures (TMs) implemented by means of information protection and organizational measures (OMs) implemented by regime measures. The combination of OMs and TMs is a way to protect information. At the same time, the contribution of these types of information security measures (ISMs) to the protection process is not always symmetrical.

The existing methodological approaches do not contain a proven mathematical apparatus that determines one or another degree of subjectivity of the decisions made, the quality of which depends on the completeness of the initial data on the current situation, the complexity of the IS structure, and the level of training of experts. This situation establishes the need to create new, and improve existing, methodological and instrumental tools to support and automate the process of making expert decisions in the tasks of developing and evaluating the effectiveness of a set of IS protection measures.

From the point of view of the development of IS, supporting decision makers helps them in building possible scenarios based on the specified algorithms and databases. As a result, government officials make more reliable decisions, even in a dynamically changing environment, and officials of governing bodies make decisions that are more verifiable and symmetrical to the current situation.

Analysis of the current state of research on the development of decision-support information systems (DSISs) [10–12] allowed us to highlight the principle of such work, which is based on the following actions:

(a). formation of initial data and intelligent analytics of the conditions within which decisions are made;

(b). situational analysis, design, and elaboration of the possible alternatives for achieving the desired goals;

(c). simulation and cognitive modeling, together with the derivation of an algorithm of actions;

(d). adaptation of the chosen solution to the emerging conditions, together with the construction of logical chains based on precedents.

According to the structural hierarchy, DSISs consist of a user interface, databases, knowledge bases, and modeling tools that implement the databases and knowledge bases. The systems themselves are classified into three main groups:

1. Passive groups when a neural network processes data, providing the user with structured information and reports; specific decisions are made by individuals.

2. Active groups that form potential solutions based on the processed database and offer possible alternative options for action.

3. Combined groups that offer possible solutions and alternatives, but at the same time allow the decision-maker to make clarifications, add conditions, and send the project for re-processing. In such situations, various models are determined, making it possible to reach the optimal decisions.

In the field of research, the third option is most acceptable.

Analysis of the scientific research and the capabilities of existing software products for modeling complex systems with probabilistic indicators [13–15] allowed us to identify the following functions necessary to perform the task of decision support in the field of information security.

The modeling function provides the process of developing solutions based on the construction of probabilistic, situational–functional, and analytical simulation and other models.

The database processing function (the information and calculation function) provides for the receipt and correction of source data based on digital repositories of information on a specific organization, its assets, IS, classes of violators, ISTs, methods and scenarios of their implementation, vulnerabilities, OMs and TMs, data of instrumental monitoring, audits of IS security during its operation, etc.

We propose to base the implementation of these two functions on self-learning artificial intelligence (S-LAI), based on the NBM. An analysis of scientific research was carried out in this area. Its results showed that in a number of works, the authors expressed the following concerns, and a number of shortcomings were noted.

In [16], aspects of the introduction of IBM Watson into the healthcare system in China were studied; the authors concluded that officials cannot control the learning process and respond to problems that arise, because the work of these officials is carried out on the principle of a "black box";

Another study [17] concluded that the operations of modern algorithms in the form of a "black box" led to the absence of an algorithm for the control of the learning process and, as a result, to a loss of confidence in the decisions taken;

In [18], the authors came to the conclusion that these algorithms performed their tasks brilliantly in theory, but in practice, for the most part, a number of serious problems arose with their training and functioning. The reason for this was insufficient volume of knowledge bases.

The knowledge-based processing function (the cognitive function) implements the software and hardware process of forming solutions in the conditions of fuzzy source data, based on structural and functional models and scenarios for performing similar or typical tasks, taking into account statistical patterns, dependencies, techniques, and algorithms, as well as making necessary adjustments for officials of management bodies in the field of information security (OISMB s) and information security.

The function of ensuring communication provides the possibility of interaction of several OISMBs working on the same task, as well as their parallel solutions for particular problems.

The documentation function provides the process for developing a set of documents on the decisions taken.

## 3. Research Methods

During the analysis of the existing modeling methods, four methods were selected as most suitable for modeling IS, taking into account their functioning processes in the conditions of complex impacts (CIs). These four methods are an ontology method for the formation of situational structural and functional models [19], the method of Bayesian networks for the formation of an NBM [20,21], and the multicriteria assessment (MCDM) methods, which are the methods of nearest neighbors and fuzzy sets used to ensure the required degree of reliability of the results of NBM training [22–24].

Ontologies are used for knowledge organization systems in those areas where it is necessary to detect infrastructure integration, to identify hidden relationships between elements (for example, structural and functional relationships of system elements and expert systems). The main postulate of an ontology is that if some objects or the connections between them are missing in the knowledge base, this does not mean that they do not exist, but simply that they are not described.

An ontology can be represented as a graph, the vertices of which are entities (concepts) and the edges of which are the relationships between entities. If any statements can be presented in the form of simple sentences, then data on the entities (concepts) mentioned in such statements and the relationships between them can be extracted. There are two main tools: the resource description framework (RDF) or ontology web language (OWL). OWL allows the description of logical rules over data. Ontologies are well suited for searching for

specific information, and knowledge bases are needed where new knowledge needs to be identified—for example, in decision-support systems (expert systems). Depending on the goals and objectives, an ontological knowledge space can be built that includes modules of subsystems, with a reflection of the processes implemented by them [25,26].

In the field of information security in relation to this study, this property of ontologies can be implemented to build an ontological knowledge space that includes ontologies of information systems with the required degrees of detail, ontologies of objects of destructive impact and their vulnerabilities, ontologies of risks of the implementation of ISTs, methods and scenarios of the implementation of ISTs, and ontologies of protective measures.

Important advantages of ontologies are their visibility; their ability to build a structural and functional model of ISs of any complexity; their ability to define impact objects, paths (routes), and the implementation of IST scenarios within IS (tactics and techniques); their ability to identify, analyze, and assess incident risks; and their methods and points of neutralization of ISTs and/or vulnerabilities.

For the full-fledged operation of the model, a probabilistic assessment of the occurrence of interrelated information security events is necessary, such as the assessment of an incident risk when implementing an IST scenario with a certain probability. Such a probability is called conditional; that is, it occurs for one event, provided that another event (according to a confirmed or unconfirmed statement) has already occurred. To calculate such probabilities, the Bayes theorem is used, the essence of which is described by the following formula:

$$P(A \mid B) P(B) = P(B \mid A) P(A) \tag{1}$$

The graphical model of this dependence is a Bayesian network of trust—a directed acyclic graph in which there is no directed route starting and ending at the same vertex [27].

The vertex of the network is either a discrete random variable with a finite number of states or a continuous Gaussian quantity.

Vertices can represent variables of any type; they can be weighted parameters, hidden variables, or hypotheses. In this study, the concepts of the ontological model are accepted by these vertices, and the functional connections are the edges of the graph and, accordingly, acquire the values of event-conditional probabilities. Thus, the study identified the property of symmetry of the developed Bayesian and ontological models of information systems, the impacts of violators, and protective measures.

To create an NBM, we applied a graphical model—a probabilistic machine learning algorithm based on the application of the Bayes theorem. When using that algorithm, it is assumed that the continuous values of all characteristics have a Gaussian distribution (a normal distribution). In this case, the distribution of numerical variables was used, with an approximate estimate of the frequency close to the true one.

The main advantages of the Bayesian Gauss algorithm include the availability of implementation and fairly low computational costs when training an NBM. If the features are independent, then the Bayesian algorithm is considered optimal. Additional advantages of this algorithm are the small amount of training information required to determine indicators; the ability to process missing values of concepts; the algorithm's speed; and the algorithm's ability to work with unknown variables. This algorithm allows the combination of patterns derived from databases and values obtained during expert evaluation.

To train the NBM considered in the work, the AutoAI artificial intelligence training algorithm proposed in the study was applied [18]. This algorithm significantly reduces the existing disadvantages of the S-LAI and has the following advantages:

1. It implements a quantitative "causal comparative analysis" based on synthesized training data on information security events that have already occurred (a priori events);
2. It implements a quantitative "correlation study", where the statistical relationship between a priori and a posteriori IS events is evaluated and their mutual influence and actual risks are determined.

3. The AutoAI algorithm can predict actual losses, including a priori and a posteriori risks of losses from an IS event.
4. The training scenario constructs use standard open-source analytics (OSINT) to collect publicly available data, including publicly available repositories (databases).
5. The AutoAI algorithm and the FAIR method (i.e., the fair resource allocation method from game theory) are integrated, using Bayesian optimization as a probabilistic iterative algorithm based on a Gaussian process or a graph model and a function for collecting training data in the areas of "exploration" and "exploitation".
6. The AutoAI algorithm can identify and adapt modern technologies, which allows its integration into complex systems to ensure their reliable cybersecurity.

Because of these advantages, the AutoAI artificial intelligence training algorithm, according to its structural and functional characteristics, can be adapted to ensure the security of information systems for various purposes.

The "nearest neighbor" method is the simplest metric algorithm based on the evaluation of the similarity of objects, according to which the classified object belongs to the class within which the objects closest to it belong to the training sample. The "nearest neighbor" method belongs to a class of methods based on storing data in memory for comparison with new elements. When a new record appears for prediction, deviations are found between this record and similar data sets, and the most similar (or nearest neighbor) is identified. The advantage of this method is its simplicity of implementation [28].

The advantages of the fuzzy logic method are its abilities to formalize and aggregate the experience of developers in setting up control loops. At the same time, a simplified method of managing complex processes is used and experience in managing processes of a certain type is preserved, taking into account exceptions of various kinds and features of the system as a whole. Accounting, the combination of different source data of different grades, and the creation of abstracted templates (classes) for detecting attacks on system resources are performed [29].

In a number of scientific papers, a new hybrid modeling, PROMETHEE-SAPEVO-M1, which is based on multi-criteria methods, has been proposed as a methodology for decision-making analysis [30]. It is implemented on the basis of integrating two methods—the PROMETHEE method [31] and the SAPEVO-M method [32], which is the evolution of the SAPEVO method [33]. The proposed methodological approach makes it possible to carry out a detailed quantitative and qualitative assessment of arrays of source data to structure a format for calculating weights for criteria for evaluating the preference of criteria and alternatives [34].

The most valuable attribute of this approach is the possibility of reducing criteria using factor analysis.

This hybrid model includes three integrated-results analysis models.

These models and techniques have a software implementation [35] developed in Python, which provides information and analytical support to a decision-making official in the process of analyzing and evaluating the object relative to the required criteria.

Next, we investigated the practical application of the selected modeling methods to determine the structural and functional relationships of information security events in IS and their probabilistic values.

## 4. Results and Discussion

We considered a formalized description of the cognitive model (CM) of the functioning of IS for various purposes in the conditions of CIs (Figure 1). The developed CM has three functional levels.

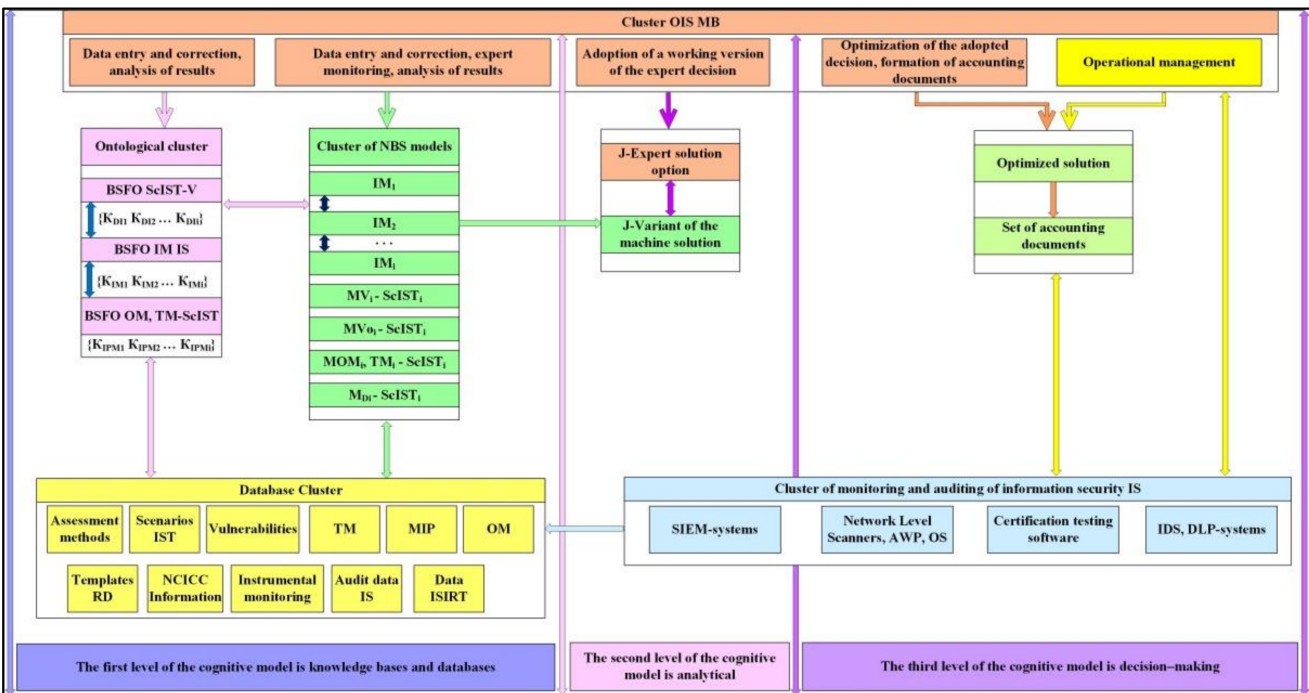

**Figure 1.** The structure of the complex model of the functioning and operational management of IS in the conditions of CIs.

The first level of the CM, knowledge bases and databases, combines clusters of an integrated ontological model, NBM, databases, and the control actions of the OISMB of the first level.

The cluster of the integrated ontological structural and functional model reflects the processes of functioning of ISs, CIs, and protective measures. The input of the initial data for its construction is carried out by the OISMB from the databases (DB) cluster.

The block of structural and functional ontologies of information modules (BSFO IM) is responsible for modeling IS (its elements). We considered the order of its functioning and application.

Structurally, the IS consists of a certain number of information modules (IMs) for various purposes. The following types of IMs are defined in this work. The first IM is represented by local information and computing networks (LICNs). They are of the second type—data processing centers, and the third type—remote users. The structure of the IS is determined by the number of types of IM and the compositions of their elements, which ensure the possibility of universality and scalability in the modeling.

In order to determine the structural and functional ontologies of the IMs of the IS as part of the block of potential CI objects, the concepts of seven levels corresponding to the levels of the ISO/OSI model were identified: physical, channel, network, transport, session, representative, and application. In addition, we included user levels. This approach made it possible to identify functional processes between the IM concepts of the LICN type at various levels (Figure 2).

Accordingly, during the construction of this LICN ontology, it was possible to determine the functional relationships between concepts of different levels, as well as to determine the physical and logical interfaces of objects to which (Table 1) CIs can be carried out; that is, the determination of the routes for the implementation of IST scenarios.

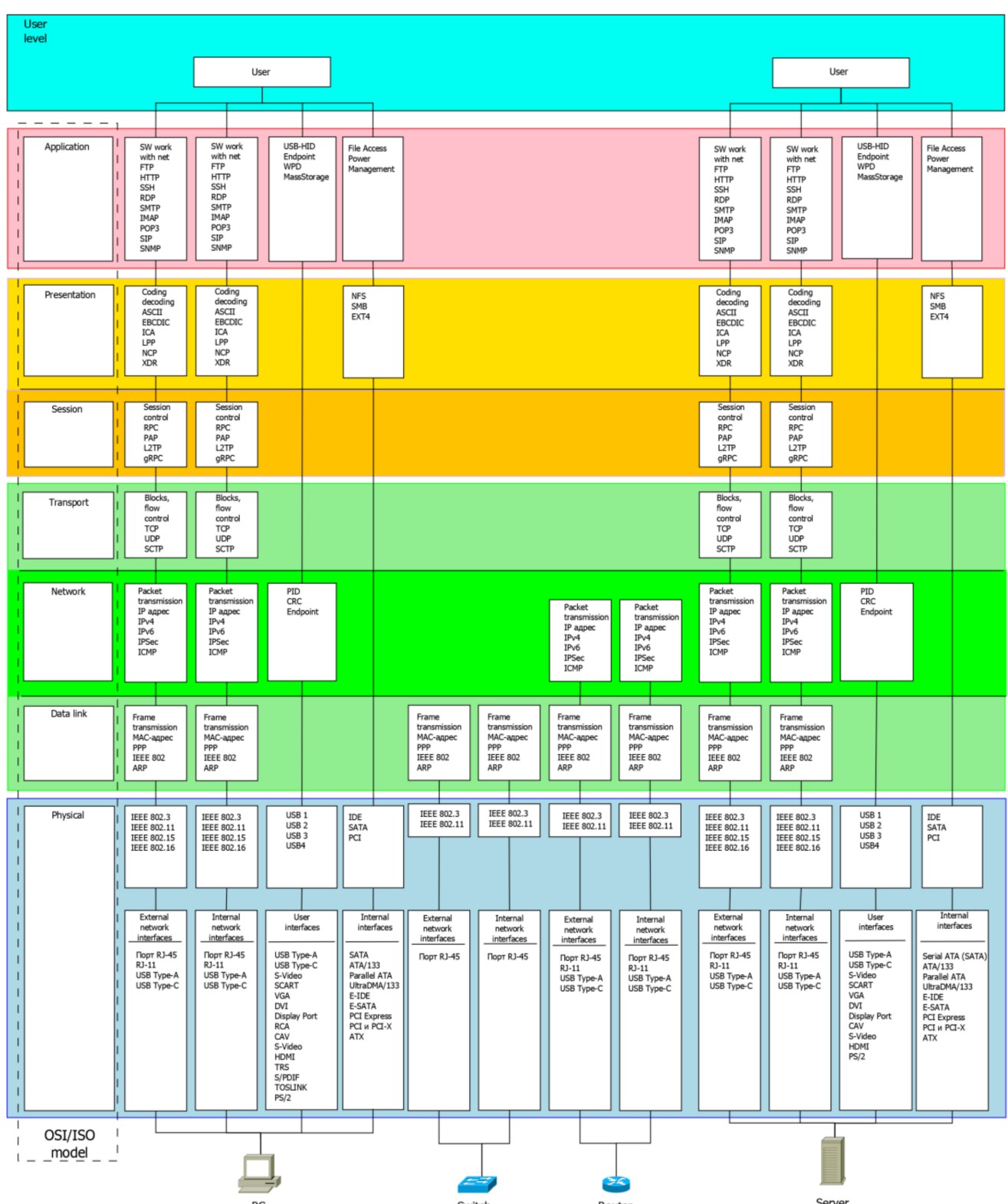

**Figure 2.** CI objects and functional connections at the levels of the ISO/OSI model for IM-1 of the LICN type.

**Table 1.** CI objects by levels of the OSI/ISO model.

| Model Levels OSI/ISO | Objects of Destructive Influence |
| --- | --- |
| User level | User data (identification, authentication, personal, corporate), information resources. |
| Application level | Application software (office software, electronic document management system, browsers, modeling software, calculation software, etc.). |
| Representative level | Operating systems, system software, system libraries, virtualization platforms, software to support protocols. |
| Session level | Operating systems, system software, system libraries, virtualization platforms, software to support protocols. |
| Transport level | Operating systems, system software, system libraries, virtualization platforms, software to support protocols. |
| Network level | Transmitted data (informational, service, and technical). (Message packets) |
| Channel level | Transmitted data (informational, service, and technical). |
| Physical level | Hardware of a personal computer, servers, data storage systems, switching and routing equipment, peripheral and network devices, elements of telecommunication systems, communication channels, and data transmission medium. |

The modeling of this process was carried out using a block of structural and functional ontologies of scenarios for the implementation of ISTs and vulnerabilities (BSFO Sc IST-V). The modeling reflects the interrelationships of the following concepts: a set of classes of violators {IC1...ICi}, ISTs, methods of implementing ISTs {M1...Mi}, a set of tactics for implementing ISTs {T1 ... Ti}, a number of techniques for implementing ISTs {t1...ti}, and a number of vulnerabilities of CI objects at the hardware, system, application, network, and user levels {V1 ... Vi} IM (ontological relationships with BSFO IM IS).

Each method has its own implementation scenario (tactics ($T_i$) and techniques ($t_i$)) through existing vulnerabilities ($E_i$). Thus, it is possible to compile an anthology of mutual influences of vulnerabilities, methods, and scenarios for the implementation of ISTs.

A fragment of the initial data illustrating its construction in the form of a matrix is shown in Figure 3.

In this matrix, the names and descriptions of the ISTs are placed horizontally, and the vulnerabilities are placed vertically. If a threat can be implemented through one or several vulnerabilities, then the name and content of the method of its implementation, $S_i$, as well as the scenario (tactics and techniques) of application, are indicated in the cells at their intersection.

The block of structural and functional ontologies of protective measures, vulnerabilities, methods of implementing protective measures, and the scenarios for the implementation of ISTs (BSFO OM, TM-V-Sc IST) models the interrelationships of the following concepts: many ways of using TMs {$TM_1$... $TM_i$} implemented by many ways of using information security tools, and many ways of using OMs {$OM_1$ ... $OM_i$} implemented. There are many ways to apply regime, organizational, and engineering measures aimed at closing the vulnerabilities of CI objects on hardware, system, application, network, and user levels {$V_1$ ... $V_i$} IM and/or localization of a variety of ways to implement ISTs {$M_1$...$M_i$}, their tactics {$T_1$ ... $T_i$}, and the techniques {$t_1$...$t_i$}.

| Name of information security threats | Threat of physical incapacitation of storage facilities | The threat of formatting media information | The threat of "forced web browsing" | Threat of theft of storage and processing facilities |
|---|---|---|---|---|
| **Description of the information security threat** | The implementation of this threat is possible provided that the violator receives physical access to information carriers (external, removable and internal drives), information processing facilities (processor, device controllers, etc.) and information input/output facilities (keyboard, etc.) | The threat lies in the possibility of loss of information stored on a formatted medium, often without the possibility of its recovery, due to deliberate or accidental execution of the formatting procedure of the media | The threat lies in the possibility of the violator gaining access to protected information, performing privileged operations or performing other destructive actions on incorrectly protected components of web applications. This threat is caused by weaknesses (or lack of) a mechanism for verifying the correctness of the data entered on web servers. The implementation of this threat is possible provided that the successful implementation of "manual entry" into the address bar of the web browser of certain web page addresses and the implementation of a forced transition through the tree of the website to pages that are not explicitly referenced on the website | The implementation of this threat is possible provided that the violator has physical access to information carriers (external, removable and internal drives), information processing facilities (processor, device controllers, etc.) and information input/output facilities (keyboard, etc.) |

**Interfaces**

| Interface | Threat of physical incapacitation | The threat of formatting media | The threat of "forced web browsing" | Threat of theft |
|---|---|---|---|---|
| external network interfaces that provide interaction with the Internet, adjacent (interacting) systems or networks (wired, b/wired, web interfaces, etc.) | | | $V_2$ $M_2$ $T_1$ $t_1 t_4 t_5$ / $V_3$ $M_1$ $T_3$ $t_1 t_2 t_3$ / $V_1$ $M_1, M_4$ $T_3, T_2$ $t_3 t_7$ / $V_4$ $M_2$ $T_1$ $t_2 t_4 t_6$ | |
| internal network interfaces that provide interaction (including through intermediate components) with components of systems and networks that have | | | $V_1$ $M_2$ $T_1$ $t_1 t_7$ / $V_3$ $M_3, M_5$ $T_2, T_3$ $t_2 t_3 t_6$ | $V_1$ $M_2$ $T_1$ $t_1 t_7$ / $V_3$ $M_3, M_5$ $T_2, T_3$ $t_2 t_3 t_6 t_7$ |
| user interfaces (wired, wireless, web interfaces, remote access interfaces, etc.) | $V_1$ $M_2$ $T_1$ $t_1 t_4$ / $V_3$ $M_3, M_5$ $T_2, T_3$ $t_3 t_5$ / $V_7$ $M_1, M_4$ $T_1, T_2$ $t_4 t_5$ | $V_1$ $M_2$ $T_3$ $t_1 t_{12}$ / $V_3$ $M_1, M_5$ $T_5, T_9$ $t_5 t_7$ | $V_1$ $M_2, M_7$ $T_7$ $t_1 t_9$ / $V_3$ $M_3, M_3$ $M_5$ $T_2, T_8$ $t_4 t_5$ | $V_1$ $M_2$ $T_1$ $t_1 t_4$ / $V_3$ $M_3, M_4$ $T_2, T_4$ $t_3 t_5$ |

**Figure 3.** Matrix form of structural and functional relationships of vulnerabilities, methods, and scenarios of IST implementation.

The process of forming ontological connections of contrasting protective measures for the ISTs is presented in Figure 4 in the form of a matrix, in which the IPMs are represented horizontally and the ISTs are represented vertically. In the cells at the intersection of the ISTs and the OMs and TMs opposed to them, lists of the information security tools implementing them and regime measures are indicated.

| | | | IPM₁ TM | IPM₁ OM | IPM₂ TM | IPM₂ OM | IPM₃ TM | IPM₃ OM | IPMᵢ TM | IPMᵢ OM |
|---|---|---|---|---|---|---|---|---|---|---|
| IST₁ | $M_i$ $M_2$ $M_1$ | $P(M_i)$ $P(M_2)$ $P(M_1)$ $T_i$ $T_2$ $T_1$ | | | | | | | List IST | Content ODD |
| IST₂ | $M_i$ $M_3$ $M_2$ $M_1$ | $P(M_i)$ $P(M_3)$ $P(M_2)$ $P(M_1)$ $T_i$ $T_3$ $T_2$ $T_1$ | | | List IST | Content ODD | | | | |
| IST₃ | $M_i$ $M_1$ | $P(M_i)$ $P(M_1)$ $T_i$ $T_1$ | | | | | List IST | Content ODD | List IST | Content ODD |
| ISTᵢ | $M_i$ | $P(M_i)$ $T_i$ | List IST | Content ODD | | | | | | |

**Figure 4.** The process of forming ontological connections of contrasting information protective measures (IPMs) with threats to information security (ISTs).

The considered ontological structural and functional blocks of the model are mutually integrated and are designed to conduct situational analysis and to identify parameters and significant factors that determine the characteristics of the CI vector on objects of seven levels of IS, points and routes of influence used by OMs and TMs, the interrelationships of information security events, and the degree of their mutual influence.

The ontological cluster is algorithmically linked to the cluster of neuro-Bayesian models (NBMs).

The basis of NBMs is composed of standard modules reflecting the probabilistic characteristics of the process of functioning of the protected IS under CI conditions (Figure 5).

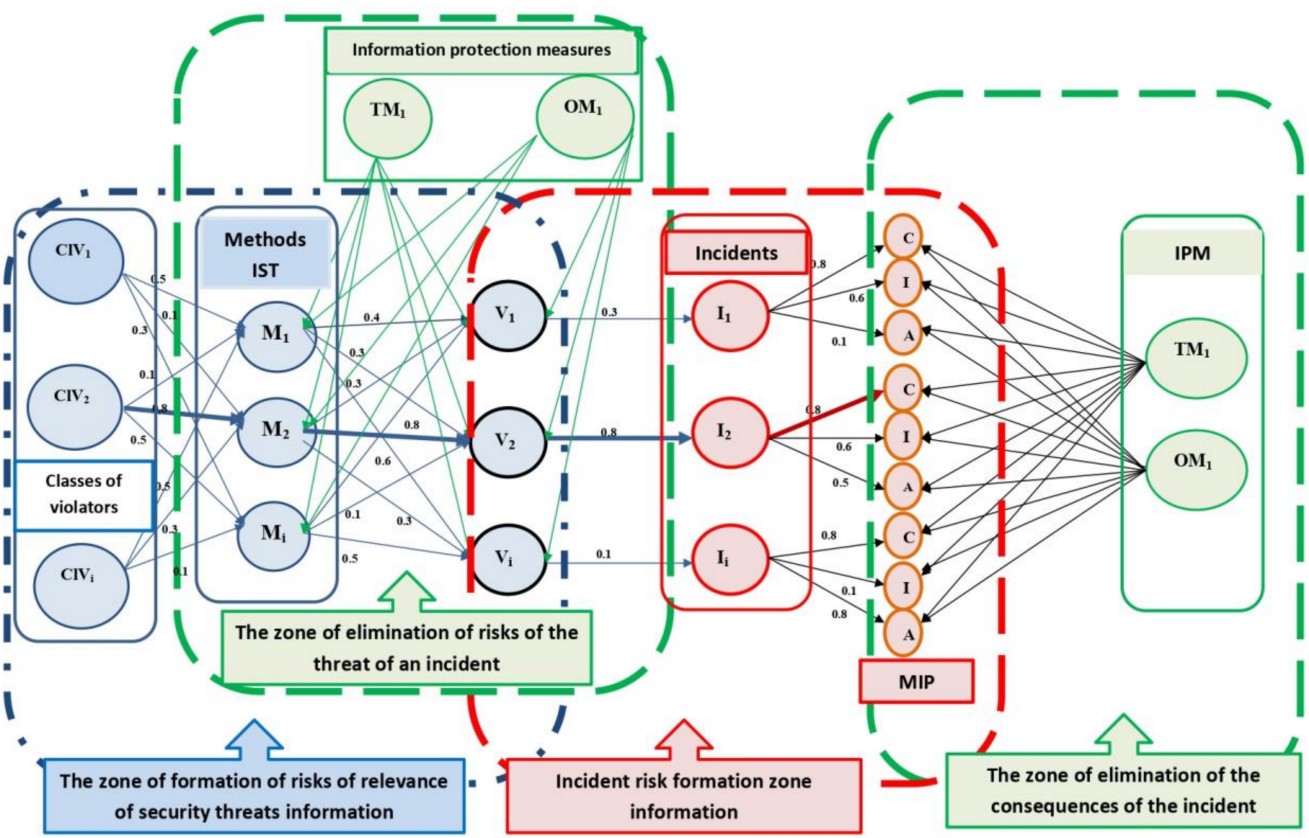

**Figure 5.** A typical NBM module for building protection of IS.

This model, based on a Bayesian network, represents a Gaussian distribution of conditional probabilities using a directed acyclic graph in which each edge is a conditional dependency and each node is a separate random variable reflecting the events of the information security. Nodes correspond to the concepts of ontological models, and edges correspond to their functional connections. Applying the symmetry property of Bayesian and ontological models, we assumed that the nodes of the Bayesian network corresponded to the concepts of ontological models and the edges corresponded to their functional connections. These properties allowed the NBM to perform the task of calculating the conditional probabilities of IS events, structurally reflected by OS-FM.

Each typical NBM module reflects the IS events associated with the risk of one IST implementation, by exploiting the set of vulnerabilities of the affected object (the OS-FM concept) in any way and its localization through the use of OMs and TMs. Two outcomes are considered: the IST was localized, and/or the UBI was implemented and led to an information security incident.

As part of a typical NBM module, four zones of information security events were allocated:

1. The zone of formation of the risks of the implementation of ISTs.
2. The zone of elimination of risks of the threat of the implementation of ISTs.

3. The incident risk formation zone.
4. The zone of elimination of the consequences of the incident.

The structure of typical NBM modules is symmetrically superimposed on the generated OS-FM of the ontological cluster. The input of the initial data is carried out in the direction of CIs and the direction of protective measures, in accordance with the developed OS-FM. The weighting coefficients of information security events are calculated using the PROMETHEE-SAPEVO-M1 release 3.0 software product.

The risk zone of information security threats is formed on the basis of the BSFO IM and BSFO interrelationships of IST implementation scenarios and the vulnerabilities of impact objects.

The probability of choosing a specific method of implementing the IST will depend on the capabilities of the violator, the required resources, the value of the protected information resources, the infrastructure characteristics of the object, its protection system, and the vulnerabilities [36].

The zone of elimination of the risks of the threat of the implementation of ISTs is formed on the basis of the BSFO interrelationships of protective measures, vulnerabilities, methods, tactics, and techniques of the implementation of ISTs.

The use of the NBM AutoAI learning algorithm allows the calculation of the probabilistic characteristics of the relevance of the ISTs and the methods and scenarios for their implementation, taking into account the mutual influence of a priori and a posteriori events of the IS, as well as selecting the necessary OMs and TMs for their closure. A fragment of this process is shown in Figure 6.

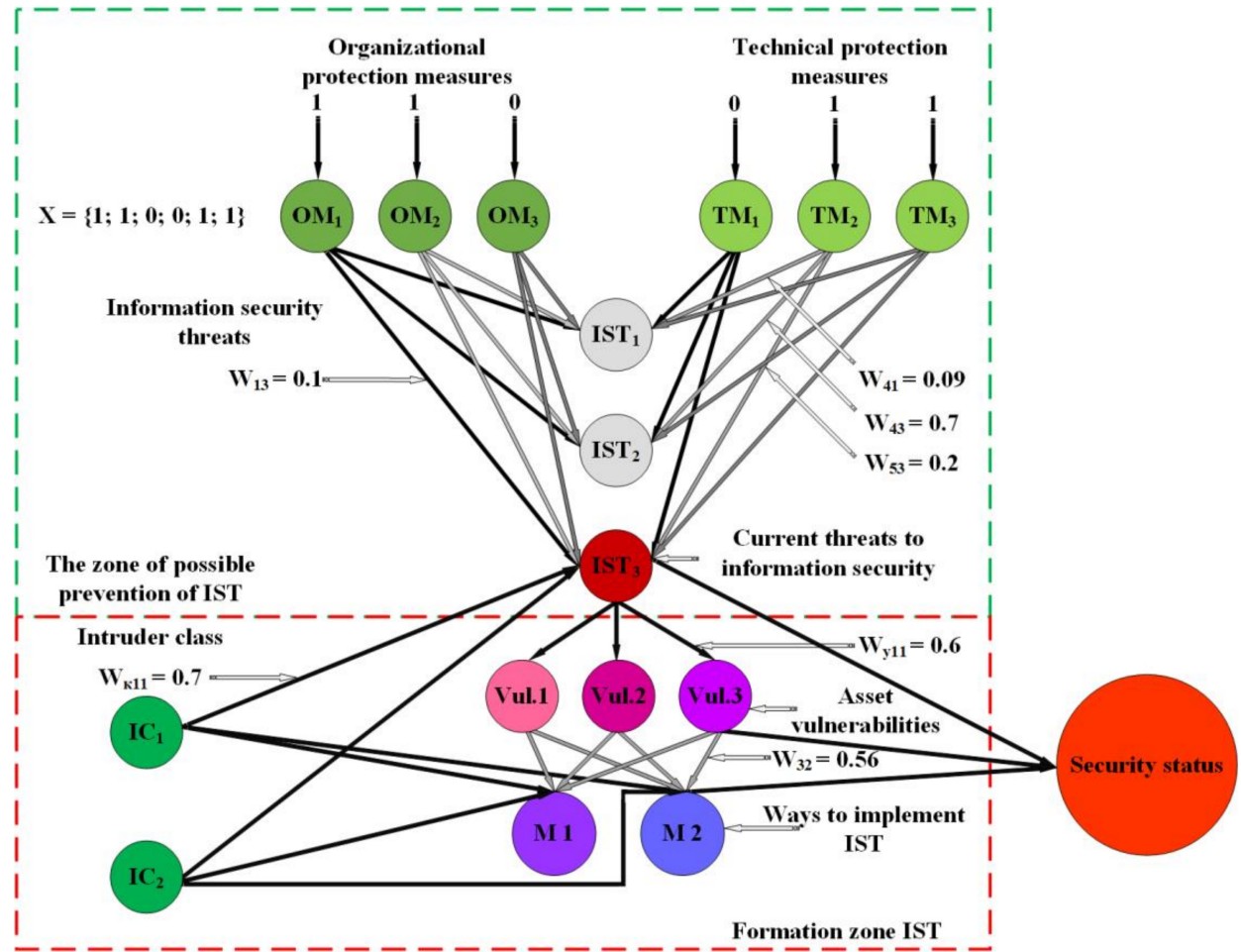

**Figure 6.** The work of the NBM in determining the probabilistic characteristics of its security.

The security criteria are determined by organizational, guidance, and methodological documents. The methods of implementing the ISM will have a probabilistic degree of effectiveness, depending on the functional characteristics of the TMs and the OMs and the functional indicators of the method of implementing the ISTs, as well as on the emergence of new vulnerabilities, ISTs, and the methods and scenarios for their implementation.

Thus, a machine version of the decision-making on the actualization of the ISTs and the methods of implementing the OMs and the TMs is created, which at the analytical level of the cognitive model is compared with the solution developed by the expert method.

The machine and expert decision-making options are symmetrical in their structure and were compared for completeness, quality, and adequacy of decision-making. Their comparative assessment was carried out according to the following indicators:

1. The types and numbers of actual ISTs.
2. The degree of neutralization of the risks of IST implementation by the applied OMs and TMs.
3. The evaluation of the effectiveness of the use of resources (i.e., the number of types of means of information protection is minimal ($Ns_T$ min), the maximum number of information security tools belongs to one vendor, and the cost of selling the IPM is less than the cost of the protected assets).
4. The achievement of the required controllability values for the elements of the information security tools.
5. The provision of the required values of intelligence protection indicators, functional and structural stability, and recoverability.

After comparison, analysis, and evaluation, the decision-making option was optimized (the third level of the cognitive model). At the same level, a set of accounting documents was formed and a simulation of the operational management system of a protected IS was carried out in the conditions of CIs.

The monitoring and auditing of IS information security in the course of operational management was carried out by an instrumental method using intrusion detection systems, Security information and event management systems, Data Leak Prevention systems, security scanners, certification testing software, as well as the method of assessing the security of IS by modeling malicious attacks (penetration testing) [37,38]. A variant of the operational management model of the protected IS is presented in Figure 7.

The data obtained were used to build up or change the structure of the protection system when applying new ISTs and methods of their implementation, or new vulnerabilities. The results of the instrumental security control were evaluated by the OISMB and changes were made to the knowledge base and database level models.

The zone of elimination of the consequences of an incident is modeled similarly to the zone of elimination of risks of threats to the implementation of incidents, with the preparation of an appropriate model.

In the standard NBM module, the incident risk formation zone is responsible for modeling the process of operational management of information security events. It takes into account the probability of damage to the confidentiality (*C*), integrity (*I*), and availability (*A*) of protected information resources with a certain degree of probability. To do this, a probabilistic model of the dependencies of the type and degree of damage on the type of incident is formed on the basis of the matrix. The parameters of damage assessment for each type of information resource are indicated horizontally. The name and content of the incident are indicated vertically. At the intersection, a verbal description of the damage and the probability of its infliction in the event of an incident is indicated [39].

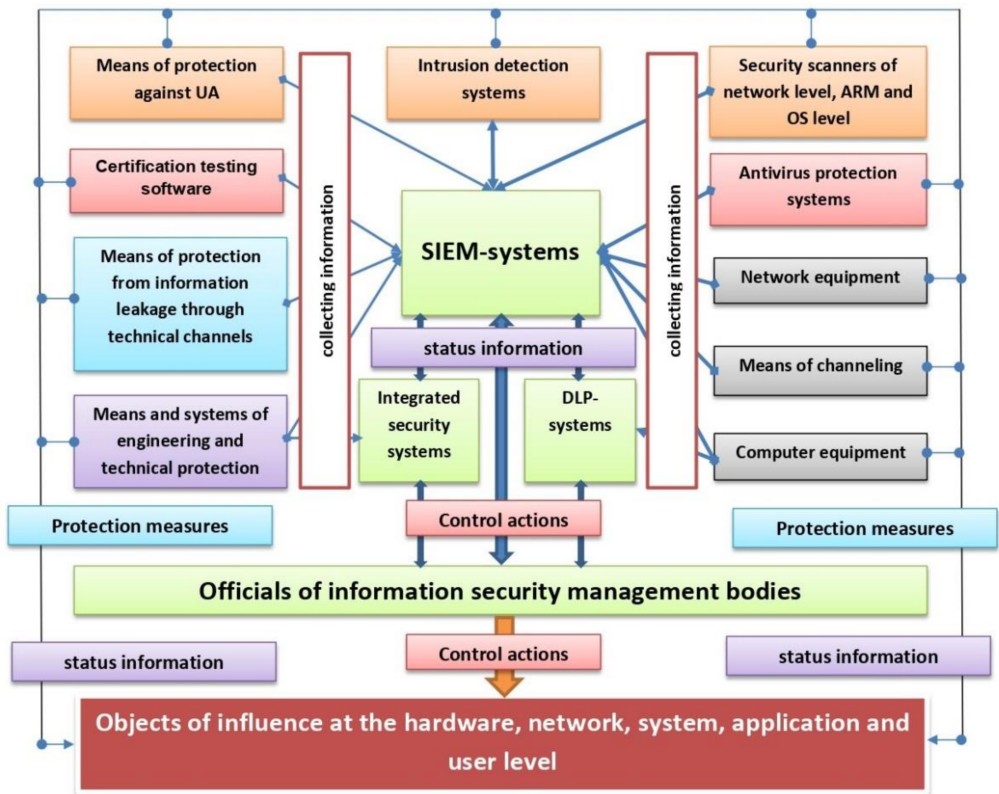

**Figure 7.** Model of operational management of information security events (options).

## 5. Methodology and Algorithms for Determining the Relevance of Scenarios and Tactics

The technique used in this article made it possible to carry out a universal assessment for information security events based on the quantitative values of a subset of the variables of the IPM and the destructive impact of violators, thereby minimizing the likelihood of an erroneous decision.

We considered the algorithm of its application for the previously developed ontological structural and functional model of IS.

Each IS consists of information modules of different compositions and structures. They may be denoted by the set $\{G_1 \ldots G_i\}$. In each IM, seven levels of structure are allocated to determine the objects of destructive influence. This may be imagined as the following tuple:

$$G_i \in \{F_i, K_i, L_i, T_i, S_i, A_i, P_i, H_i\} \tag{2}$$

where $F_i$ represents the physical, $K_i$ represents the channel, $L_i$ represents the network, $T_i$ represents transport, $S_i$ represents session, $A_i$ stands for representative, $P_i$ represents application, and $H_i$ stands for user levels of the $i$-th IM.

In the model, potential objects of destructive influence at these levels are represented by concepts interconnected by functional connections.

Each of these objects has a specific set of vulnerabilities $\{V_1 \ldots V_i\}$. The vulnerabilities are determined during penetration testing, or monitoring and auditing of information security. We denoted these activities by the index $T_i$, where $i$ is the type of testing.

For each information module, we introduced the concept of a tuple of its vulnerabilities at each of the levels, as follows:

$$G_i \in \left\{\sum_1^n V_{Fi}, \sum_1^n V_{ki}, \sum_1^n V_{Li}, \sum_1^n V_{Ti}, \sum_1^n V_{Si}, \sum_1^n V_{Ai}, \sum_1^n V_{Pi}, \sum_1^n V_{Hi}\right\} \tag{3}$$

To ensure the binding of this vector to the ontological model, we compiled the following matrices of vulnerabilities of concepts of each level. The matrix columns display

concepts $\{K_{1Ai} \ldots K_{mAi}\}$ of the corresponding level and, in the lines, the detected vulnerabilities of each of concept ((4a)–(4h)):

$$
G_{iF_i} = \begin{bmatrix} K_{1Fi}V_1 & K_{1Fi}V_2 \ldots & K_{1Fi}V_n \\ K_{2Fi}V_1 & K_{2Fi}V_2 \ldots & K_{2Fi}V_n \\ K_{3Fi}V_1 & K_{3Fi}V_2 \ldots & K_{3Fi}V_n \\ K_{mFi}V_1 & K_{mFi}V_2 \ldots & K_{mFi}V_n \end{bmatrix}
\tag{4a}
$$

$$
G_{iK_i} = \begin{bmatrix} K_{1Ki}V_1 & K_{1Ki}V_2 \ldots & K_{1Ki}V_n \\ K_{2Ki}V_1 & K_{2Ki}V_2 \ldots & K_{2Ki}V_n \\ K_{3Ki}V_1 & K_{3Ki}V_2 \ldots & K_{3Ki}V_n \\ K_{mKi}V_1 & K_{mKi}V_2 \ldots & K_{mKi}V_n \end{bmatrix}
\tag{4b}
$$

$$
G_{iL_i} = \begin{bmatrix} K_{1Li}V_1 & K_{1Li}V_2 \ldots & K_{1Li}V_n \\ K_{2Li}V_1 & K_{2Li}V_2 \ldots & K_{2Li}V_n \\ K_{3Li}V_1 & K_{3Li}V_2 \ldots & K_{3Li}V_n \\ K_{mLi}V_1 & K_{mLi}V_2 \ldots & K_{mLi}V_n \end{bmatrix}
\tag{4c}
$$

$$
G_{iT_i} = \begin{bmatrix} K_{1Ti}V_1 & K_{1Ti}V_2 \ldots & K_{1Ti}V_n \\ K_{2Ti}V_1 & K_{2Ti}V_2 \ldots & K_{2Ti}V_n \\ K_{3Ti}V_1 & K_{3Ti}V_2 \ldots & K_{3Ti}V_n \\ K_{mTi}V_1 & K_{mTi}V_2 \ldots & K_{mTi}V_n \end{bmatrix}
\tag{4d}
$$

$$
G_{iS_i} = \begin{bmatrix} K_{1Si}V_1 & K_{1Si}V_2 \ldots & K_{1Si}V_n \\ K_{2Si}V_1 & K_{2Si}V_2 \ldots & K_{2Si}V_n \\ K_{3Si}V_1 & K_{3Si}V_2 \ldots & K_{3Si}V_n \\ K_{mSi}V_1 & K_{mSi}V_2 \ldots & K_{mSi}V_n \end{bmatrix}
\tag{4e}
$$

$$
G_{iA_i} = \begin{bmatrix} K_{1Ai}V_1 & K_{1Ai}V_2 \ldots & K_{1Ai}V_n \\ K_{2Ai}V_1 & K_{2Ai}V_2 \ldots & K_{2Ai}V_n \\ K_{3Ai}V_1 & K_{3Ai}V_2 \ldots & K_{3Ai}V_n \\ K_{mAi}V_1 & K_{mAi}V_2 \ldots & K_{mAi}V_n \end{bmatrix}
\tag{4f}
$$

$$
G_{iP_i} = \begin{bmatrix} K_{1Pi}V_1 & K_{1Pi}V_2 \ldots & K_{1Pi}V_n \\ K_{2Pi}V_1 & K_{2Pi}V_2 \ldots & K_{2Pi}V_n \\ K_{3Pi}V_1 & K_{3Pi}V_2 \ldots & K_{3Pi}V_n \\ K_{mPi}V_1 & K_{mPi}V_2 \ldots & K_{mPi}V_n \end{bmatrix}
\tag{4g}
$$

$$
G_{iH_i} = \begin{bmatrix} K_{1Hi}V_1 & K_{1Hi}V_2 \ldots & K_{1Hi}V_n \\ K_{2Hi}V_1 & K_{2Hi}V_2 \ldots & K_{2Hi}V_n \\ K_{3Hi}V_1 & K_{3Hi}V_2 \ldots & K_{3Hi}V_n \\ K_{mHi}V_1 & K_{mHi}V_2 \ldots & K_{mHi}V_n \end{bmatrix}
\tag{4h}
$$

where matrix (4a) represents the vulnerabilities of the physical level concepts, matrix (4b) represents the vulnerabilities of the channel level concepts, matrix (4c) represents the vulnerabilities of the network level concepts, matrix (4d) represents the vulnerabilities of the transport level concepts, matrix (4e) represents the vulnerabilities of the session level concepts, matrix (4f) represents the vulnerabilities of the representative level concepts, matrix (4g) represents the vulnerabilities of application level concepts, and matrix 4h represents the vulnerabilities of the user-level concepts.

Accordingly, we obtained a set of objects of destructive impact distributed across the levels of the information system model and their vulnerabilities, which are the points of implementation of the ISTs in any way.

Next, it was necessary to rank the concepts by assigning them weights of influence on the survivability of the information module. In particular, we introduced a matrix of vectors of critical significance of concepts of each level of IM, in the following form:

$$W_{G_i} = \begin{bmatrix} w_{1K_{Fi}} & w_{2K_{Fi}} & w_{3K_{Fi}} \cdots & w_{mK_{Fi}} \\ w_{1K_{Ki}} & w_{2K_{Ki}} & w_{3K_{Ki}} \cdots & w_{mK_{Ki}} \\ w_{1K_{Li}} & w_{2K_{Li}} & w_{3K_{Li}} \cdots & w_{mK_{Li}} \\ w_{1K_{Ti}} & w_{2K_{Ti}} & w_{3K_{Ti}} \cdots & w_{mK_{Ti}} \\ w_{1K_{Si}} & w_{2K_{Si}} & w_{3K_{Si}} \cdots & w_{mK_{Si}} \\ w_{1K_{Ai}} & w_{2K_{Ai}} & w_{3K_{Ai}} \cdots & w_{mK_{Ai}} \\ w_{1K_{Pi}} & w_{2K_{Pi}} & w_{3K_{Pi}} \cdots & w_{mK_{Pi}} \\ w_{1K_{Hi}} & w_{2K_{Hi}} & w_{3K_{Hi}} \cdots & w_{mK_{Hi}} \end{bmatrix} \tag{5}$$

The weight (significance) of the $i$th concept was determined for the survivability of the IM as a whole. The weighting coefficients were determined by the expert method from the OS-FM analysis.

The values of the weights varied from 0 to 1. The critical significance of the weights was determined to be in the range from 0.8 to 1.

The following rule may be considered: "If a vulnerability is discovered during testing, then it must be protected by the use of technical means or organizational measures." To implement this rule, we formed a matrix of vectors of protective measures using the initial data from the previously formed BSFO of the interrelationships of protective measures, vulnerabilities, methods, tactics, and techniques for the implementation of ISTs.

To implement this, we carried out the following activities.

We updated IST threats and scenarios of their implementation, using the NBM cluster.

We updated the list of concepts available for destructive impact and their vulnerabilities ((4a)–(4h)). These links are registered in databases.

We created a matrix for the distribution of information security equipment and ongoing regime measures. These links are registered in databases.

We connected the NBM cluster and ed data on the protective measures that were carried out.

We attended to the output of a report and ensured that all vulnerabilities were closed and/or the risks of implementing IST scenarios were localized.

However, our experience suggested that there are no absolutely protected IS, due to the following reasons:

1. Not all vulnerabilities were uncovered during testing;
2. The computer intelligence of the violators applied a new method of opening objects of influence;
3. The violators developed a new attack scenario that allows the circumvention of the applied information security tools;
4. The violators destroyed (blocked) the applied information security tools;
5. The information security tools failed during operation, or their parameters were configured incorrectly;
6. The violators gained physical access to the elements of the IS and/or to the information security system.

To determine the potential for destructive effects on the concepts, it was necessary to carry out the following actions:

For each information module, we introduced the concept of a vector of probabilities of detecting its vulnerabilities by the violators' computer intelligence, as follows:

$$G_i \in \{p_{V1}, p_{V2}, \ldots, p_{Vn}\} \tag{6}$$

We established the following matrices:

the probability matrices of opening vulnerabilities ($P_{AUT.Vi}$), opening of the elements of the protection system ($P_{AUT.ST}$);

the probability matrices of survival of the $i$th information module ($P_{sur.IMi}$) in case of physical, software impact and exploitation of vulnerabilities of concepts;

a matrix of probabilities of the survival of elements of the protection system ($P_{sur.ST}$) in case of physical and (or) programmatic impact;

matrices of probabilities of maintaining operability when exposed to intentional or unintentional radio–electronic interference on the means of protection ($P_{REI.ST}$) and IM elements ($P_{REI.EIMi}$); and

a matrix of probabilities of proper functioning of protective equipment ($P_{rel.\ ST}$) and IM elements ($P_{rel.\ EIMi}$) during the operation in conditions of critical changes in parameters, including erroneous actions of personnel.

The calculation of the probabilistic characteristics mentioned above should include the weights of the criticality of the concepts of IM at all its levels [40].

The property of symmetry of the Bayesian and ontological models, noted earlier in this study, made it possible to develop a method for calculating the above probabilistic characteristics of the destructive impact using the NBM. The approach proposed allows updating the probabilities of the information security events reflected in the models whenever new information becomes available. The mathematical basis for this is the Bayes theorem.

With a correctly compiled model and reliable information about information security events, it can be proved that the methodology embedded in the model ensures the correct calculation of updated probabilities, relative to the axioms of classical probability.

Any node (concept–vulnerability) in different network clusters can receive information, because the method does not distinguish between an inference in the direction of edges or vice versa. The simultaneous input of information into several nodes will not affect the algorithm as a whole.

In the typical module of the model (Figure 5), the mutual influence of destructive influences and IPMs are presented in the form of chains of consecutive events and convergent and divergent sequences of events.

Events in successive chains can be homogeneous (destructive or protective) and heterogeneous (destructive and protective) (as shown in Figure 8).

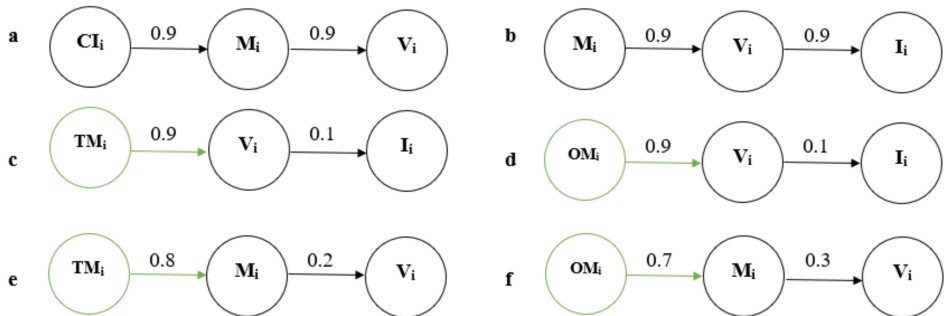

**Figure 8.** Types of sequential chains of events: (**a,b**)—chains of homogeneous (destructive) events, (**c–f**)—heterogeneous sequences of events.

The calculation of the joint probability distribution for both consecutive homogeneous and heterogeneous events is carried out according to the following formulae:

For option Figure 8a—chains of homogeneous (destructive) events:

$$P(IC_i, V_i | S_i) = \frac{P(IC_i, M_i, V_i)}{P(M_i)} = \frac{P(IC_i)(M_i | IC_i)P(V_i | M_i)}{P(M_i)} = P(IC_i | M_i)P(V_i | M_i) \quad (7)$$

Here, the a priori probability that the violator of the $IC_i$ will be able to implement the ISTs by the $M_i$ method will depend on the a posteriori probability of the presence of a vulnerability that is not closed for this method of $V_i$.

For option Figure 8c—chains of heterogeneous events:

$$P(TM_i, I_i | V_i) = \frac{P(TM_i, V_i, I_i)}{P(V_i)} = \frac{P(TM_i)(V_i | TM_i)P(I_i | V_i)}{P(V_i)} = P(TM_i | V_i)P(I_i | V_i) \quad (8)$$

Here, the a priori probability of the effectiveness of technical measures will depend on the a posteriori probability of the presence of an uncovered vulnerability.

Similarly, the formulae are compiled and the probabilistic dependences of information security events are established for other types of chains of consecutive events.

The model also identified the mutual influence of destructive influences and IPMs in the form of chains of convergent events (Figure 9).

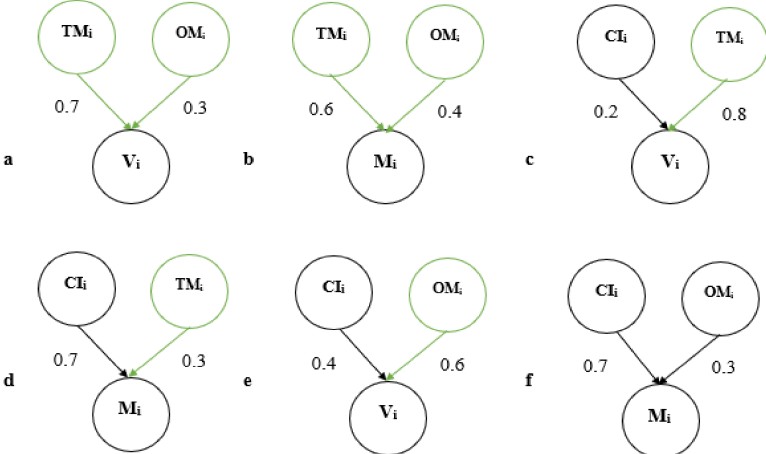

**Figure 9.** Types of convergent event chains. (**a,b,f**)—chains of homogeneous events, (**c,d,e**) — heterogeneous sequences of events.

The joint probability distribution for option Figure 9a of the convergent sequence of information security events is calculated using the following formula:

$$P(TM_i, OM_i, V_i) = \sum_{V_i} P(TM_i)P(OM_i)P(V_i|OM_i, TM_i) = P(OM_i|V_i)P(TM_i|V_i) \quad (9)$$

For divergent events, the a priori probability of the existence of vulnerability $V_i$ will depend on the a posteriori probabilities of its closure with the help of $OM_i$ and $TM_i$.

Similarly, the formulae are compiled and the probabilistic dependencies of information security events are established for other types of chains of convergent events.

The mutual influence of destructive influences and IPMs in the form of chains of divergent events is shown in Figure 10.

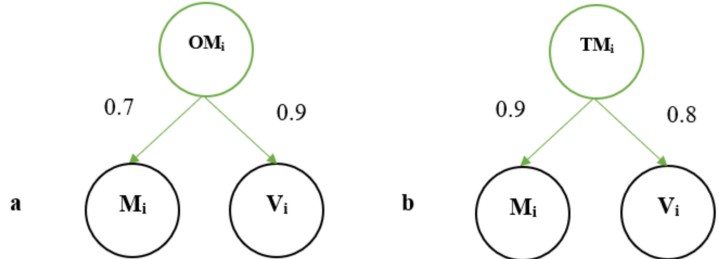

**Figure 10.** Types of divergent event chains. (**a,b**)—chains of homogeneous events.

The joint probability distribution for a convergent sequence of information security events is calculated using the following formula:

$$P(M_i, V_i|TM_i) = \frac{P(TM_i, M_i, V_i)}{P(TM_i)} = P(S_i|TM_i)P(V_i|TM_i) \quad (10)$$

For divergent events, the a priori probability of the existence of vulnerability $V_i$ and the possibility of implementing ISTs by the $M_i$ method will depend on the a posteriori probabilities of the implementation of $OM_i$ and/or $TM_i$.

This approach allows us to calculate the dependence of the a priori probability of events on the probability of events that may occur; that is, to determine the probabilistic dependence of destructive events on protective ones, and vice versa.

## 6. Conclusions

The cognitive model of the functioning of IS for various purposes, as presented in this work in the conditions of CIs, is the next step in the development of the theory and methodology of information security in terms of creating new, and improving existing, methods, models, and algorithms for decision-making support in the formation of a set of measures to protect IS for various purposes.

The first scientific contribution was the proposed approach related to the decomposition of elements of a distributed information system at the levels of the OSI/ISO model and the definition of objects (concepts) of destructive impact at each of these levels. This made it possible to identify the vulnerabilities of each concept, identify entry points and ways of spreading information security threats, and, consequently, form vectors of tactics and scenarios for their implementation.

The next scientific contribution was the development of a three-level cognitive model. Its core is an ontological structural and a functional model of a distributed information system under conditions of destructive influence, integrated with a probabilistic NBM. This integration of two different types of models became possible because of the symmetry properties of the structure of the elements of the IS, the processes they implement, and the events of information security of various origins as identified during this study. These properties made it possible to obtain a new quality of the integrated model, with the ability to evaluate a priori and a posteriori dependences of the probabilities of proper functioning of processes, in a distributed information system, on the implementation of information security events of various origins.

The standard information modules and the standard modules of information security events, as developed and applied in the model, provided the properties of universality and scalability of the cognitive model.

The next scientific contribution was the possibility realized by the model of leveling the danger of incorrect decision-making when decisions are formed by artificial intelligence or during expert decision-making. This contribution was made by a symmetrical comparison of the expert version with the machine version, together with subsequent optimization.

The proposed methods made it possible to determine the degree of criticality for the IS of objects of destructive influence, the vulnerability of the IS, attack vectors, and vectors of protective measures. As a result, we obtained a technological map of the security status of the ICs or their elements.

The primary calculation of the probabilities of destructive and protective information security events (i.e., their weights) were performed using the PROMETHEE-SAPEVO-M1 method and a software product.

In the dynamics of operational management, their recalculation was carried out using Bayes formulae for sequential, converging, and diverging chains of events. The applied learning algorithm of S-LAS AutoAI made it possible to calculate the probability of a priori and a posteriori events of IS.

Our review of scientific research confirmed that the models and methods considered in this article were presented to the scientific community for the first time.

Further research in this area will include the development of techniques that implement formalized algorithms for determining scenarios, tactics, and techniques for implementing ISTs and determining their preferences in the current situation. Other interesting research areas will be the development of databases and knowledge bases for teaching NBM using the AutoAI algorithm, the expansion of the scope of the methodology. and the PROMETHEE-SAPEVO-M1 software product in the developed DSIS.

The practical significance of the research results lies in their use in the activities of organizations carrying out IS certifications. The application of the research results can

significantly reduce the time of work on the formation of the IST model, the development and selection of OMs and TMs, regime measures, increases in the indicators of the validity of decisions made, and the reliability of the results of assessments of the security of IS for various purposes.

**Author Contributions:** Conceptualization, funding acquisition—A.A.S.; formal analysis, methodology, validation, visualization, writing—original draft—V.V.B. and A.A.S. All authors have read and agreed to the published version of the manuscript.

**Funding:** The work was carried out with the financial support of an MTUSI grant provided by the Ministry of Finance of the Russian Federation from the federal budget in 2021 (scientific project No. 35/21-d) within the framework of the federal project "Information Security" of the national program "Digital Economy of the Russian Federation".

**Data Availability Statement:** Not applicable.

**Conflicts of Interest:** The authors declare no conflict of interest. The funders had no role in the design of the study; in the collection, analyses, or interpretation of data; in the writing of the manuscript; or in the decision to publish the results.

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
