# Peer review of "Cognitive Model for Assessing the Security of Information Systems for Various Purposes"

_symmetry, doi:10.3390/sym14122631_

Round 1

Reviewer 1 Report

The work proposes developing a cognitive model for assessing the security of information systems. The paper is interesting, well-written, and structured. As suggestions for improvement, I highlight the points below:

The introduction presents sentences that need references to support what is being stated. The text has several short paragraphs, which makes the reading less fluid.

Considering the relevance of the topic and the impact of this journal, the Literature Review should be extended, presenting recent studies on Artificial Intelligence and Mathematical Modeling, citing studies that present approaches similar to those presented in this paper, such as:

https://ieeexplore.ieee.org/document/9810236

http://dx.doi.org/10.1590/0103-6513.20210011

https://doi.org/10.1142/S0219622021500565

Also, the authors fail to highlight trends and publication gaps on the topic. Thus, authors should highlight the main contributions of their article to the academic literature, compared with the articles already published.

Finally, the results and conclusions must be improved, explaining the paper's main contributions to the scientific community and society.

Author Response

Dear colleague, the authors of the article want to express their deep gratitude for such a high assessment of their work.
We have carefully studied the content of your comments and generally agree with them.
We have made a number of changes in the text of the article. In the introduction, links have been added to confirm the stated statements and proposals. The text design has been adjusted by combining paragraphs logically. The section "Review of scientific literature on the research topic" was added and the literature review was expanded. You have kindly provided links to very interesting studies containing new methodological approaches in the field of mathematical modeling and artificial intelligence. This, of course, allowed us to expand and more reasonably present the main provisions and conclusions contained in our article. The article also more clearly defined the current problems and trends in this field of science and how they are reflected and solved in the presented material. The conclusions were adjusted in the direction of concretization of the results obtained.
We wish you health, creative success and hope for fruitful cooperation and are waiting for a response.

Kind regards, doctor Vladimir Baranov, doctor Alexander A. Shelupanov

Department of "Information Security" M.I. Platov South Russian State Polytechnic University, Novocherkassk, Russia.

Reviewer 2 Report

Accept as it is.

The method and model are significant and valuable. However, the content and organization of this paper can be improved.

1. Please modify the key words, about 3-5 words.

2. The author's introduction needs to be optimized. What is the main contributions of this paper? How is the paper structured?

3. Please make sure that every symbol has a paraphrase.

4. What are the limitations of this study?

5. What are the further research topics and directions? What else can be improved? My suggestion is that you would add this part in conclusion.

Author Response

Dear colleague, the authors of the article want to express their deep gratitude for such a high assessment of their work.
We have carefully studied the content of your comments and generally agree with them.
We have made a number of changes in the text of the article.
1. Keywords, roughly abbreviated.
2. The introduction reveals what is the main contribution of research to science and what is the structure of the article. In addition, the bibliographic list on the research topic was expanded and the section "Review of scientific literature on the research topic" was added. Restrictions are also specified. accepted in the study.
3. All meanings of symbols and abbreviations are checked and disclosed.
In conclusion, more specifically, with an emphasis on the contribution to science, the conclusions were adjusted and the topics and directions of further research were identified.
We wish you health, creative success and hope for fruitful cooperation and are waiting for a response.

Kind regards, doctor Vladimir Baranov, doctor Alexander A. Shelupanov

Department of "Information Security" M.I. Platov South Russian State Polytechnic University, Novocherkassk, Russia.

Reviewer 3 Report

Very interesting ant timely article. I think it deserves publication and I am recommending accept with minor corrections. But there are some minor issues that require your attention. I list these corrections below as feedback / comments, and I am looking forward to reading the updated version of this article. 

The article is well structured and well written. I think it deserves a consideration for publication. There are some minor corrections which I outline in more detail in the 'comments for authors' section.

- You have done a really good job at reviewing so many articles, but you didn’t discuss future developments on AI. There is a recent articles on this topic that review recent and relevant literature, ‘the ‘future values and risks from artificial intelligence’ - see: https://doi.org/10.1007/s12553-022-00691-6 - It would be interesting to see a few sentences reviewing and comparing your work in relations to this recent study in a related topic. 

Well done on the paper, it's a very interesting text. 

Author Response

Dear colleague, the authors of the article want to express their deep gratitude for such a high assessment of their work.
We have carefully studied the content of your comments and generally agree with them.
. You have kindly provided a link to very interesting studies containing new methodological approaches in the field of methods for collecting and processing databases and knowledge bases for training artificial intelligence. The AutoAI algorithm developed by you is universal and can be perfectly integrated into the cognitive model developed by us, because according to the above description, it corresponds to structural and functional indicators in everything. It would be very interesting to adapt it for an information decision support system in the field of information security. This was indicated in the text of the article and the corresponding links were made.
I would like to get acquainted in more detail with the mathematical, software and structural solutions implemented in the algorithm you have developed. We are ready and hope for cooperation in this scientific field.
We wish you health, creative success and hope for fruitful cooperation and are waiting for a response.

Kind regards, doctor Vladimir Baranov, doctor Alexander A. Shelupanov

Department of "Information Security" M.I. Platov South Russian State Polytechnic University, Novocherkassk, Russia.

Round 2

Reviewer 1 Report

The authors made the proposed changes and raised the level of the article. Therefore, I suggest accepting the paper.